# Highly Sensitive Piezoresistive Pressure Sensor Based on Super-Elastic 3D Buckling Carbon Nanofibers for Human Physiological Signals’ Monitoring

**DOI:** 10.3390/nano12152522

**Published:** 2022-07-22

**Authors:** Zhoujun Pang, Yu Zhao, Ningqi Luo, Dihu Chen, Min Chen

**Affiliations:** 1State Key Laboratory of Optoelectronic Materials and Technologies, School of Electronics and Information Technology, Sun Yat-sen University, Guangzhou 510275, China; pangzhj3@163.com; 2School of Physics, Sun Yat-sen University, Guangzhou 510275, China; stscm@mail.sysu.edu.cn; 3School of Materials and Energy, Guangdong University of Technology, Guangzhou 510006, China; zhaoyu@gdut.edu.cn (Y.Z.); nqluo@link.cuhk.edu.hk (N.L.)

**Keywords:** 3D buckling carbon nanofibers, mechanical property, piezoresistive sensitivity, flexible piezoresistive sensor

## Abstract

The three-dimensional (3D) carbon nanostructures/foams are commonly used as active materials for the high-performance flexible piezoresistive sensors due to their superior properties. However, the intrinsic brittleness and poor sensing properties of monolithic carbon material still limits its application. Rational design of the microstructure is an attractive approach to achieve piezoresistive material with superior mechanical and sensing properties, simultaneously. Herein, we introduce novel three-dimensional buckling carbon nanofibers (3D BCNFs) that feature a unique serpentine-buckling microstructure. The obtained 3D BCNFs exhibit superior mechanical properties, including super-elasticity (recovery speed up to 950 mm s^−1^), excellent flexibility (multiple folds), high compressibility (compressed by 90%), and high fatigue resistance (10,000 bending cycles). The pressure sensor fabricated by the 3D BCNFs shows a high sensitivity of 714.4 kPa^−1^, a fast response time of 23 ms, and a broad measuring range of 120 kPa. The pressure sensor is further applied to monitor the physiological signals of humans, and is capable of detecting the characteristic pulse waves from the radial artery, fingertip artery, and human-breath, respectively.

## 1. Introduction

Electronic skin (e-skin) with high flexibility, preeminent sensitivity, and lightweight has attracted wide attention and led to various applications in artificial prosthetics, humanoid robotics, and wearable devices [1,2,3]. Several relevant transduction principles of pressure sensors were proposed in the past, including piezoresistivity [4], piezocapacitance [5], and piezoelectricity [6]. Among them, the piezoresistive sensor was studied in depth because of its static and dynamic pressure monitoring, simple structure, rapid response, and high stability. Different microstructures have been introduced in electrodes or piezoresistive material, such as pyramids [7,8], hemisphere [9], and micropillar arrays [10]. In some studies, conductive fillers were embedded in the stretchable and flexible bulk polymers [11,12,13]. To realize real-time physiological signal monitoring for patients and even healthy individuals in daily life for early warnings or diagnoses of disease, the flexible pressure sensor is highly desirable to realize high sensitivity, a wide measurement region, a fast response speed, and high mechanical flexibility, simultaneously.

Three-dimensional (3D) monolithic porous structures have become a forefront material for flexible and wearable pressure sensors due to their low density, high compressibility, and high surface area [13,14]. Many 3D carbon nanostructures, such as carbon nanotube aerogel [15,16,17], carbonaceous nanofibrous aerogels [18,19,20], nanocellulose aerogels [21,22,23], and graphene aerogel [4,24,25,26,27], have been synthesized and utilized for pressure sensors [14,28,29,30,31]. However, these conventional carbonized materials are brittle and have a lack of ductility or yielding, which would lead to large plastic deformation when being bent or under high-compression strain. Moreover, these materials commonly suffer from a poor piezoresistive effect due to the uniform high conductivity without a large resistance change per unit pressure [19,20,32]. Particularly, due to the outstanding flexibility derived from the one-dimensional longitudinal characteristic, electro-spun carbon nanofibers (CNF) are being extensively investigated as one of the alternatives in fabricating a flexible 3D structure [33,34,35]. However, these conventional electro-spun fiber mats are dense as their fibrous networks pack tightly in the perpendicular direction, resulting in poor mechanical properties of compressibility, elasticity, and flexibility [36,37]. Recently, Han et al. has introduced an electrospinning strategy containing high ionic strength salt solutions to fabricate 3D carbon nanofiber networks, based on which a pressure sensor with a sensitivity of 1.41 kPa^−1^ can be achieved [38]. Cai et al. proposed ferrosoferric oxide (Fe_3_O_4_)/carbon nanofibers fabricated by three-dimensional electrospinning and then further fabricated a pressure sensor with a working range of 0–4.9 kPa and a sensitivity of 0.545 kPa^−1^ [32]. However, these 3D electro-spun carbon nanofibers fabricated by electrostatic repulsion possess poor mechanical properties [38], since cotton-like 3D structures are loose resulting from the random accumulation of nanofibers [39,40], and the straight 1D nanostructures have mechanical compliance properties [41]. These loose 3D electro-spun carbon nanofibers inevitably lack elasticity and are prone to densification under only a relatively low applied pressure, revealing a low response speed, small measuring range, and a low sensitivity. It is a great challenge to fabricate an advanced piezoresistive carbon material that simultaneously realizes high sensitivity, a wide measuring range, and a fast response speed.

Note that, the microstructure geometrical engineering on building blocks is regarded as a promising route to improve the mechanical properties and increase the surface area [2,42]. The buckling structure is widely used in flexible electronic devices to improve the flexibility, compressibility, and elasticity of inherent hard materials [43]. The significantly altered mechanical properties of carbon materials can be obtained by constructing a unique microstructure on basic building blocks that serve as elastic units [42]. Moreover, buckling structures have been used to improve sensitivity in many reports, and because of that, buckling surfaces have substantial advantages over planar surfaces due to their larger surface area, variable surface morphology, and abundant contacting modes [44].

In this work, by geometrical engineering of the microstructure, a novel advanced 3D buckling carbon nanofiber (BCNF) material which consists of many serpentine-shaped nanofibers is fabricated. The serpentine-shaped buckling nanofibers decorated with projecting tentacle-like carbon nanotubes have excellent mechanical properties and good sensing properties, simultaneously. The 3D BCNF exhibits super-elasticity, superior flexibility, high compressibility, and fatigue resistance. It is capable of jumping five times its own height when released from bending and can automatically unfold with a fast recovery rate after multiple folding, standing out from other traditional porous carbon materials. The 3D BCNF-based sensor possesses an ultrahigh-pressure sensitivity of 714.4 kPa^−1^, a fast response time of 23 ms, a broad loaded range of 120 kPa, a low detection limit (40 Pa), and high stability (>2000 loading/unloading cycles). Finally, the sensor is successfully demonstrated for the physiological monitoring of pulse waves from the radial artery, carotid artery, fingertip artery, and human-breath, indicating that the sensor has potential applications in health monitoring.

## 2. Materials and Methods

### 2.1. Preparation of Carbon Nanofibers

Nanofiber precursors were fabricated via electrospinning. Firstly, solution A was prepared by adding 0.8 g of polyacrylonitrile (PAN) to 3.6 g of dimethylformamide (DMF) during magnetic stirring for 4 h. Solution B was prepared by adding 0.6 g of CNT to 5 g of DMF via ultrasonic dispersion over 2 h. Then, solutions A and B were mixed and magnetically stirred for 20 h to obtain the precursor solution. The precursor solution was loaded into a syringe and the feeding rate of the solution was (controlled) 2 mL h^−1^. The distance between the tip of the needle and the covered collector was about 20 cm, and a high voltage (20 kV) was applied. The relative humidity during electrospinning was below 45% and the temperature was ~30 °C. The electro-spun fibers were directly collected on a metal-foil-covered metallic rotating roller, typically performed over 2 h. Finally, the nanofibers were carbonized in a tube furnace at 900 °C in nitrogen for 2 h, and the temperature was increased with a rate of 20 °C min^−1^. The control samples of straight carbon nanofibers (SCNF) and pure carbon nanofibers (PCNF) were prepared using conventional annealing of the electro-spun carbon fibers. The content of CNT in the control groups SCNF and PCNF was 50% and 0% of the sample 3D BCNF, respectively. Prior to the carbonization in the tube furnace, the electro-spun fibers were pre-oxidized at 240 °C for 2 h to obtain pre-oxidized nanofibers. Then, the nanofibers were carbonized at 900 °C in nitrogen for 2 h, with a heating rate of 5 °C min^−1^. The PCNF sample was prepared under the same conditions.

### 2.2. Sensor Fabrication

The flexible pressure sensor was fabricated using a sandwich structure. This was achieved by assembling the 3D BCNF composite mats on top of a pair of interdigitated Au electrodes and encapsulating with a 1.0 mL acrylic PSA adhesive, followed by covering with a 1.4 mm-thick PEN film. The interdigitated Au electrodes (Au thickness = 100 nm; electrode width = 200 mm; interval = 100 mm; active pressure-sensitive area = 3 × 3 mm) were patterned on a 2.3 m-thick PI-coated Si-wafer using photolithography followed by magnetron sputtering. An anisotropic conductive wire was bonded to the interdigitated electrodes to connect them with standard DuPont pins.

### 2.3. Characterization of the Material

The morphology of the carbon nanofibers was determined using field-emission scanning electron microscopy (SEM, Zeiss/Bruker, Gemini500, Jena, Germany) and transmission electron microscopy (TEM, JEM-2100, JEOL). Raman spectroscopy was performed with a LabRAM HR 800 UV laser micro-Raman spectroscope (HORIBA Jobin Yvon, Palaiseau, France), with a laser excitation wavelength of 532 nm. The Brunauer–Emmett–Teller (BET) surface analysis was performed with a surface area porosity analyzer (BSD-66).

### 2.4. Device Characterization Platform

The pressure-sensing test was performed with an experimental setup consisting of a high-accuracy load cell (LSB200, USB200, FUTEK), a high-frequency piezoelectric actuator (NAP100, Newport), and a high-speed Keithley source-meter (Model 2636B). During the pressure-sensing test, the sensor was placed between the actuator and the load cell. The actuator was controlled to generate movement that can create pressure on the sensor, and the force was detected by the load cell. The source-meter applied a constant voltage (1 V) to the sensor and measured the output current of the sensor. Both the applied force and the current were recorded with the LabVIEW program and saved in a computer.

## 3. Results

Figure 1a illustrates the fabrication process of the 3D BCNF. Firstly, 2D polymer nanofibers were directly prepared by electrospinning the precursor solution of polyacrylonitrile (PAN) and CNT. The nanofibers aligned parallel to the plane of the collector and packed closely in a layer-by-layer manner. Immediately, without a pre-oxidation process, the 2D PAN nanofiber mat was carbonized at a large heating rate under argon atmosphere to buckle the longitudinal fibers into a 3D serpentine morphology, accompanied by a significant increase in thickness (Appendix A). For the control group, SCNF and PCNF were fabricated by a conventional treatment of slow annealing after pre-oxidation. The rapid annealing process with a high heating rate endows the nanofibers with a greater shrinkage [45]. In addition, the absent of pre-oxidation leads to the rapid evaporation of the residual solvent, which thus strengthens the shrinkage. The introduction of many embedded carbon nanotubes increased the uneven stress distribution during annealing. Therefore, through the synergistic effect of large shrinkage caused by rapid annealing and uneven stress caused by adding a large number of carbon nanotubes, the unique 3D serpentine-shaped buckling structure of carbon nanofibers was prepared.

As shown in Figure 1b, the scanning electron microscopy (SEM) images and photographs demonstrate that the 3D BCNF can be folded up to 180° without cracking. On the contrary, the SCNF (Figure 1c) and PCNF (Figure 1d) both break down when bent to a certain limited level. The red dashed circle in Figure 1b highlights the concerned region. It is seen that neither a single fracture of individual nanofibers nor a structural collapse was found, indicating the excellent flexibility of the 3D BCNF. Using high-resolution SEM, we further examined the microstructures of these carbon nanofiber mats to better understand the origin of the flexibility. The monolithic carbon nanofiber mats revealed numerous microscale 3D spring-like buckling nanofibers with elastic serpentine structure, as shown in Figure 1e,h. However, the carbon nanofibers of SCNF and PCNF in Figure 1f,g were relatively straight and densely stacked in the perpendicular direction. Fibers with a spring buckling morphology are capable of absorbing more strain than their straight counterparts [46]. The serpentine 3D BCNF was randomly distributed and unidirectional, which enabled an isotropic response and stable output of the sensor as the performance is based on the behavior of these fibers as a whole.

As shown in Figure 1i, the magnified SEM image of the buckling nanofibers clearly shows a serpentine-shaped structure with a buckling radius of ~400 nm and a fiber diameter of 200 nm. In addition, the carbon nanotubes partly protrude from the carbon fiber to form tentacle-like structures. The TEM image of Figure 1g also shows that the carbon nanotubes are partially embedded in the carbon fibers. It has been reported in some studies that the addition of CNTs can improve the toughness of carbon fibers by enhancing the order of the formed graphitic structures [47,48]. In Appendix A, the Raman spectra of 3D BCNF displays two noticeable peaks (at around 1346 and 1573 cm^−1^), which correspond to the D-band and the G-band of graphitic carbon. Moreover, the volume of the 3D BCNF with low density (25 mg cm^−3^) was clearly larger than that of the conventional straight fiber membranes after annealing (Appendix A). The sample of 3D BCNF is so light that it can even stand on a tip of soft grass (Appendix A). The Brunauer–Emmett–Teller (BET)-specific surface area of 3D BCNF demonstrates a relatively higher value (244 m^2^/g) than that of SCNF (88 m^2^/g) and PCNF (43 m^2^/g).

The superior mechanical properties of the 3D BCNF are demonstrated in two experiments. A video from a high-speed camera recorded that a piece of bent 3D BCNF can bounce back with a fast recovery speed (950 mm s^−1^) from the buckling state to its original height. The bouncing strength is so large that the sample “jumped” to a distance five times its own height (Figure 2a,b and Appendix A). The above behavior suggests the spring-like super-elastic properties of the 3D BCNF. To our knowledge, this is the first time a spring-like “jump” has been reported for a carbon nanofiber material. The recovery speed of this 3D BCNF is faster than that of many reported hard carbon nanofiber aerogels and other carbon-based graphene aerogels (Appendix A) [15,20,22,30,42,49]. The 3D BCNF mats can be folded thrice in a row and automatically unfolded immediately (~0.1 s) without any structural damage (Figure 2c and Appendix A). Figure 2d is the corresponding schematic illustration of the folding in Figure 2c. Thanks to the serpentine buckling microstructure, the 3D BCNF material overcomes the brittleness problem of conventional carbon materials and exhibits superior flexibility, suitable for various wearable applications. In addition, we have folded the material more than 10,000 times and found that it remains intact, suggesting its excellent fatigue-resistance properties.

To demonstrate the super-elastic and highly compressible properties, the mechanical properties of 3D BCNF were systematically investigated using both simulations and experiments. In our simulations, a serpentine-shaped fiber with a simple supported boundary was used as a simplified structural unit for 3D BCNF. The simulated stress–strain curves for the serpentine-shaped nanofiber and straight nanofiber with the same length and same bending angle (Figure 3a) are shown in the Figure 3b. The serpentine-shaped nanofiber was subjected to less stress than the straight fiber for a given strain. The fibers with spring buckling morphology were able to absorb more strain than their straight counterparts [46]. Large arcs of serpentine can form a larger three-dimensional space to increase the internal variable contact area, which is preferable for achieving the high-pressure sensitivity. In addition, a simulation of the bending axis in different positions or opposite bending directions also showed that the serpentine spring fibers were also subjected to less stress than the straight ones (Appendix A). The simulations of the mechanics showed that this serpentine spring model can sustain large geometric deformation compared with straight fiber when under pressure.

In experiments, the elasticity (bending in lateral directions) and compressive properties (perpendicular to the carbon material plane) were studied. Firstly, as shown in Figure 3c and Appendix A, 3D BCNF was bent by 180° over 10,000 times without accumulating any damage. Notably, the nearly linear elastic regime covered a wide range of strain (from 10% up to 80%), which derived from bending deformation of the serpentine-shaped carbon nanofibers. To determine the compressibility and elasticity of carbon nanofiber mats perpendicular to the lamellas, stress–strain curves of 3D BCNF, SCNF, and PCNF were measured. As shown in Figure 3d, a narrow hysteresis loop was observed for 3D BCNFs, which suggests that both plastic deformation and energy dissipation in the compress–release cycles of 3D BCNFs were substantially lower than in conventional ones. Three-dimensional BCNF demonstrated only a negligible decay of compressive strength (2%), slight plastic deformation (1.5%), and energy dissipation of about 30% after 10 compression cycles. The compressive strength, plastic deformation, and energy dissipation of 3D BCNF were much lower than those in SCNF and PCNF (Appendix A). The significantly altered mechanical properties of serpentine-shaped BCNF originate from the construction of the unique microstructures on basic building blocks. The crescent-shaped stress–strain curves of 3D BCNF are different from open-cell foams [50,51,52], but resemble previously reported hyper-elastic 3D graphene carbon materials with buckling microstructure [42]. The energy dissipation is most likely caused by the friction between fibers or by the movement of air through the buckling fibers [22,51]. In addition, to better estimate the enhanced elasticity of 3D BCNFs, the hysteresis loop in the stress–strain curve was determined for different compressive strains (Figure 3e). For more than 80% compression, the 3D BCNF still maintained over 90% of its maximum stress and underwent only 2% plastic deformation. When at a strain level of 60%, there was only a negligible change in both maximum stress (0.2%) and permanent deformation (0.6%) (insert image in Figure 3e). The above results indicate that 3D BCNF can tolerate high elastic deformation without significant damage accumulation or structural collapse.

The 3D BCNF was further compressed for 10^3^ cycles at a strain level of 90% to test its fatigue-resistant ability and maximum compression performance (Appendix A). After this test, the 3D BCNF still maintained over 86% of maximum stress and suffered ~10% permanent deformation. As shown in Appendix A, the static compression test indicated that the elastic strength could be maintained at a constant value, without viscoelastic-like stress-relaxation, when the sample was compressed (and held) at a 50% strain level. Due to the unique serpentine-shaped microstructure of carbon nanofibers, the 3D BCNF is endowed with excellent mechanical properties, including super-elasticity, high flexibility, fast recovery speed, low energy-loss coefficient, and fatigue resistance, which make it stand out from other state-of-the-art 3D carbon nanofiber materials.

Flexible piezoresistive sensors typically have a structure that consists of a sandwiched pressure-sensitive composite material and a vertically flexible substrate with electrodes. As shown in Figure 4a,b, the flexible sensors were fabricated by using 3D BCNF as pressure-sensitive layers. The pressure sensitivity is defined as S = (ΔI/I_0_)/ΔP, where ΔI denotes the relative current change caused by a change in resistance, I_0_ is the initial current, and ΔP is the difference in pressure load. The sensitivity ultimately depends on the change rate of contact resistance under per unit pressure. As shown in Figure 4c, the sensitivity curve for the 3D SCNF sensor revealed three clearly different linear regimes. The first regime has a low-pressure range (0–5 kPa) with the highest sensitivity of 714.4 kPa^−1^. In the subsequent moderate-pressure range (5–20 kPa), the sensor showed a sensitivity of 255.8 kPa^−1^. In the high-pressure regime (>20 kPa), the sensor exhibited a sensitivity of about 26.1 kPa^−1^. For comparison, SCNF- (Figure 4d) and PCNF-based (Figure 4e) sensors were fabricated and measured, which showed sensitivity values several orders of magnitude lower than that of 3D SCNF.

As shown in Figure 4f, the piezoresistive properties of the 3D SCNF sensor feature three deformation processes. The 3D buckling nanofiber structure endows a large, adjustable, pressure-dependent contact area in this carbon material. Before applying pressure, the contact area between the carbon nanofibers or between carbon nanofibers and electrodes was very small, corresponding to a high-resistance state. Firstly, when the pressure was in the range of 0–5 kPa, numerous protruding carbon nanotubes on the carbon fibers came into contact with each other to create highly conductive paths. This resulted in a huge change in total resistance per unit pressure, revealing the highest sensitivity. Secondly, when the pressure was increased up to 10 kPa, the contact area between the carbon nanofibers gradually increased, corresponding to a moderate sensitivity. Finally, when the pressure further increased, the fibers were closely pressed together, producing a large geometric deformation without fracture. Furthermore, the sensing performance during the three stages was quite stable and showed high repeatability (Appendix A).

However, the carbon nanofibers in traditional 2D electro-spun carbon mats are tightly stacked and in parallel contact with the electrodes, which cause lower initial resistance and a small adjustable range of body contact resistance. Therefore, 2D electro-spun carbon mats have a relatively low sensitivity and measuring range. Although some other 3D structures have been assembled by straight electro-spun carbon nanofibers, they are loose and lack elasticity, and are prone to densification when a small pressure is applied, revealing a small measuring range and low response speed [32,38]. Herein, the 3D buckling structure endowed the carbon nanofiber material with not only a large contact area, but also unprecedented mechanical properties compared with other 3D porous carbon material. The compressibility of 3D BCNF allowed the sensor to withstand high pressure and resulted in a large measuring range (0–120 kPa), as shown in Appendix A. As shown in Figure 4j, the super-elasticity endowed the sensor with a fast response time (~23.3 ms) and recovery time (~23.7 ms), which were faster than previously reported results based on pure 3D carbon materials (Figure 4k).

Figure 4g shows the current–voltage curves of the 3D SCNF sensor for different pressures, with voltages ranging from −1 to 1 V. The observed curves are consistent with Ohm’s law. We tested the repeated current response under different pressures (Figure 4h). The pressure test at 75 kPa at different cycles is shown in Appendix A. An excellent steady-sensing performance for detecting different pressure loadings and a minimal pressure detection of 40 pa were found. Furthermore, the stability and durability of the flexible sensor was tested by 2000 compression/release cycles under a pressure of 10 kPa (Figure 4i). No signal-drift nor degradation was found during the cycling tests. This suggests excellent repeatability, good stability, and a long operating life of the sensor. As summarized in Figure 4l and Appendix A, our 3D BCNF-based pressure sensor showed an incomparably high sensitivity and an ultrabroad work range of pressure, outperforming existing 3D porous carbon-based pressure sensors reported in the literature, to the best of our knowledge.

The practical application of the sensor in detecting physiological signals of humans was demonstrated. The sensor is able to monitor the arterial pulse signal by attaching the sensor to the wrist of a human subject, as illustrated in Figure 5a. The pulse waveform showed a heartrate of 75 beats per minute. The wrist pulse is a key medical index, which is closely related to the heartrate and blood pressure. As shown in Figure 5b, a magnified pulse–pressure curve exhibited three clearly distinguishable peaks, which correspond to percussion wave (P-wave), tidal wave (T-wave), and diastolic wave (D-wave), respectively. The P-wave is caused by the early systolic spike during blood ejection of the contracting left ventricle of the heart. The T-wave and D-wave are part of the descending limb, which are both caused by blood flow reflection. The radial artery augmentation index (AIr) is defined as the ratio of the amplitude of the T-wave to the P-wave, and ΔT_DVP_ is the time difference between these two peaks [6,53]. The above indicators are closely associated with arterial stiffness, which can be used to evaluate cardiovascular health conditions [53]. The calculated value of AIr for the tested human subject was 71%, with a ΔT_DVP_ of 274 ms, which suggest no vascular stiffening. However, a flexible sensor is susceptible to noise and possibly becomes ineffective due to unintentional external impact or movement of the human limbs. As shown in Figure 5c, the sensor can acquire stable pulse signals under external pressure applied by a pen, which suggests a good anti-jamming capability.

The pulse wave velocity (PWV) is a marker for arterial stiffness and is a convenient signal to estimate cardiovascular risk [54]. Calculating the PWV involves simultaneous use of sensors at two sites to determine the pulse curves. The time interval between the P-waves of these cures is defined as the pulse transmit time (PTT). Then, the distance between these two sites, divided by PTT, defines the PWV. In this case, sensors were attached to the carotid artery and the lateral epicondyle vessel, for which the distance was 0.38 m in the tested subject (Figure 5d). As depicted in Figure 5d, with an average PPT calculated as 66.5 ms, the PWV was calculated to be 5.7 ms^−1^. This value is comparable with the reported PWV (~4.5 ms^−1^) in a healthy 30-year-old person [6]. Due to the high sensitivity (as described in the previous section), our sensor can also detect the very weak fingertip pulse reliably (Figure 5e and Appendix A). Furthermore, a change of the fingertip pulse at different breathing states can also be clearly recorded. As shown in Figure 5e, when the volunteer held their breath for about 10 s during the test, the amplitude of the fingertip pulse wave was found to decline. It subsequently recovered when the person breathed normally. The T-wave showed a higher peak when volunteer held their breath, as seen in Figure 5f. In addition, ΔT_DVP_ increased from 270 (normal breathing) to 291 ms. These results show that that our sensor has a pressure sensitivity high enough to clearly detect various physiological signals, which has great potential for clinical applications.

Figure 5g shows speech recognition by detecting various responses to words with different numbers of syllables. When the subject spoke words such as: bisyllabic “sensor”, monosyllabic “Hi”, and trisyllabic “science”, corresponding characteristic peaks with different intensities for different words were found in the recording signals. As shown in Figure 5h, the flexible sensor was simply attached to the outside of an ordinary mask without compromising its integrity. Different respiration states can be detected in response to the pressure on the mask that is generated by the change of airflow during breathing. The sensor can distinguish the breathing states of normal breathing, slow and deep breathing, and fast and shallow breathing by different breathing rates (~18, 12, and 30 s^−1^, respectively). As shown in Figure 5i, an experiment was carried out to further demonstrate high sensitivity over a broad pressure regime. The sensor was first compressed under high pressure by placing a 100 g weight on top, which corresponds to a reference pressure value of 109 kPa. Then, by adding a bean on top of the weight, an effective pressure increment of ΔP ~ 204 Pa was realized. The corresponding current changes of ΔP are displayed in the magnified figure in Figure 5i, which demonstrates that a tiny pressure change can be precisely detected under high pressures. These results suggest that the pressure sensor with its high sensitivity, wide measuring range, and fast response speed has a great potential to be used in wearable healthcare applications.

## 4. Conclusions

In summary, a 3D carbon nanofiber material that consists of many buckling nanofibers with a serpentine spring-like structure has been fabricated. This 3D buckling nanofiber has improved the mechanical properties of electro-spun carbon nanofibers, revealing a super-elasticity, high compressibility, good flexibility, and good fatigue resistance. The material can “jump” in a spring-like fashion and unfold with a fast recovery rate without any cracking. The flexible sensor based on this 3D carbon material has shown superior piezoresistive properties, with a sensitivity of 714.4 kPa^−1^, response time of 23 ms, and pressure measurement range of 120 kPa, demonstrating a best performance among the existing carbon nanofiber materials. The sensor has been successfully used to monitor physiological signals of the human body (e.g., pulse waves from the radial artery, fingertip artery, and human-breath state). The geometrical engineering of the microstructure design could provide guidelines for improving the mechanical properties of porous materials. Although only a small fraction of the many potential practical applications was explored, the 3D BCNF, with its excellent sensing properties, can likely be used in many areas, such as smart skins, electrodes of energy-storage devices, and so on.

## Figures and Tables

**Figure 1 nanomaterials-12-02522-f001:**
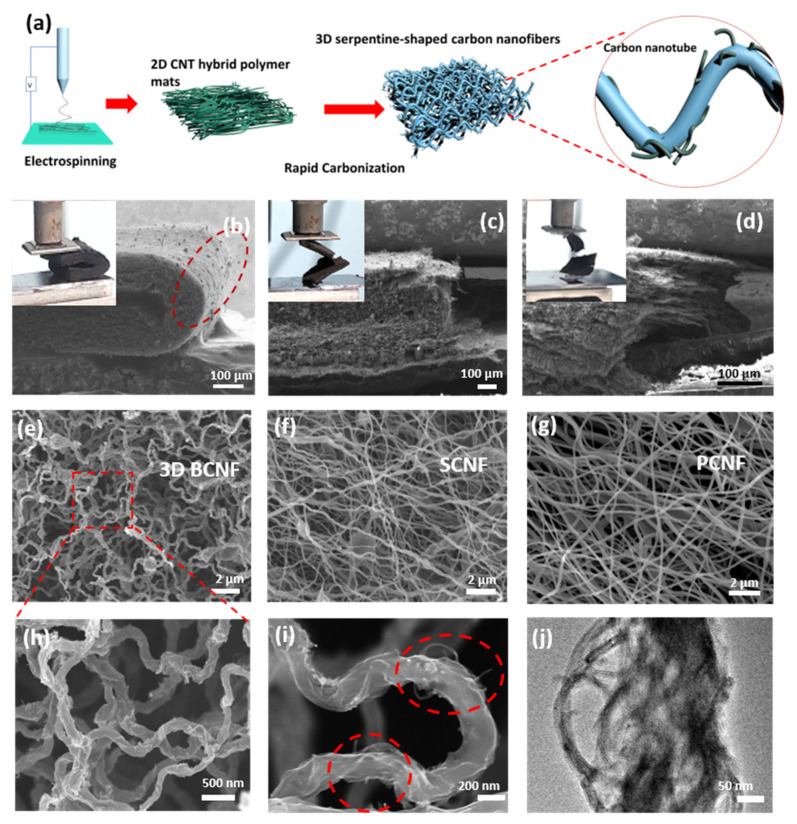
(**a**) Schematic diagram for the preparation process of 3D BCNF. Photographs and SEM images to demonstrate that (**b**) 3D BCNF can be folded up to 180° without fracture, while (**c**) SCNF and (**d**) PCNF are broken. SEM images of (**e**) 3D BCNF, (**f**) SCNF, and (**g**) PCNF. (**h**,**i**) High-resolution SEM image of 3D BCNF. (**j**) TEM image of 3D BCNF.

**Figure 2 nanomaterials-12-02522-f002:**
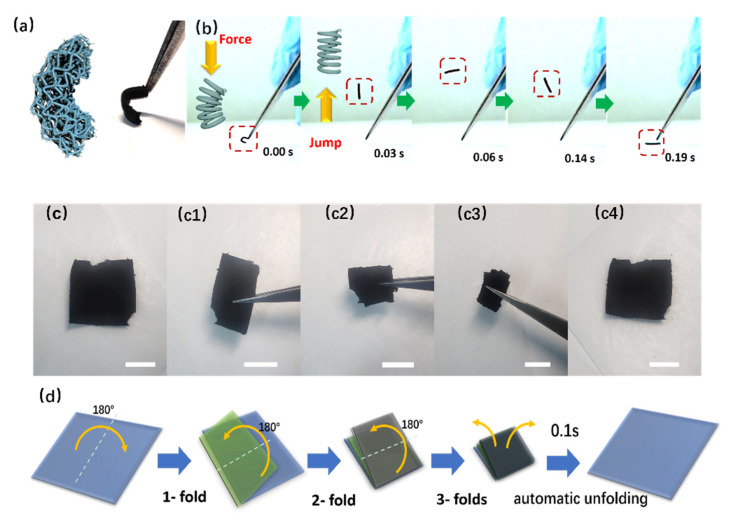
(**a**) Schematic illustration and optical images of bending 3D BCNF. (**b**) Real-time images from a high-speed camera showing that 3D BCNF can jump like a spring when released from bending. (**c**) Real-time images from the camera. (**c1**–**c4**) 3D BCNF mat being folded three times and automatically unfolding. (**d**) Schematic illustration of the process of the 3D BCNF mat being folded three times and automatically unfolding.

**Figure 3 nanomaterials-12-02522-f003:**
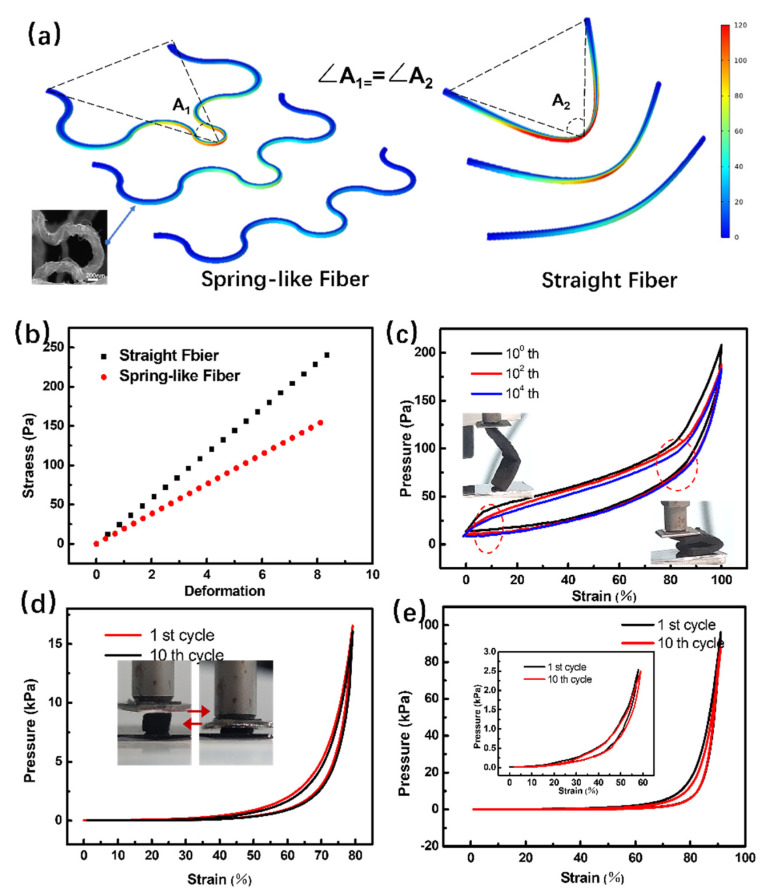
(**a**) Mechanical simulations of the material strain profiles of straight and spring-like fibers under different geometry deformations. (**b**) Simulated stress–strain curve based on straight and spring-like fibers. (**c**) Stress–strain curves of the bend–release cycle. Stress–strain curves of the compress–release cycle under a compression strain of (**d**) 80% and (**e**) over 90% strain.

**Figure 4 nanomaterials-12-02522-f004:**
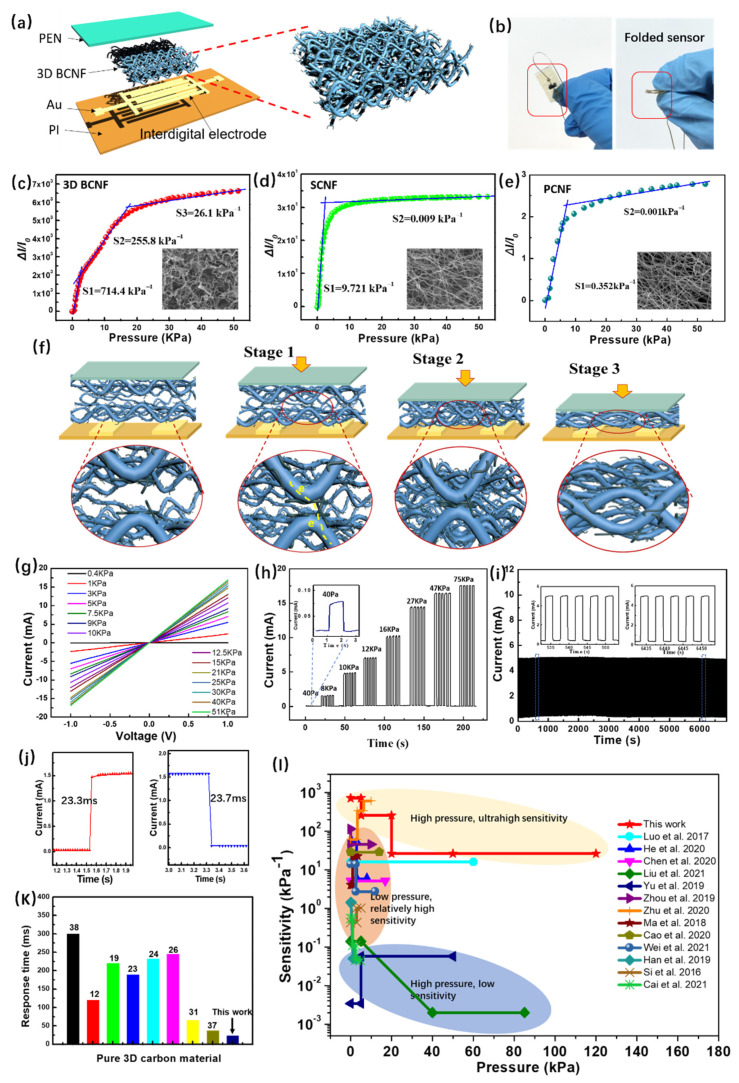
(**a**) Schematic illustration of the flexible piezoresistive sensors with a sandwich structure. (**b**) Photographs of sensor at unfolding and folding. Relative current change ratios as a function of the pressure for (**c**) 3D BCNF, (**d**) SCNF, and (**e**) PCNF. (**f**) Schematic diagram of the deformation and hierarchical conduction mechanism of the flexible 3D BCNF sensor. (**g**) Current–voltage curves under different pressures. (**h**) Repeat responding current under different pressures. (**i**) Cycling stability test of 3D BCNF under a repeated applied pressure of 10 kPa for 2000 cycles. (**j**) Response time and recovery time of 3D BCNF sensor [12,19,23,24,26,31,37,38]. (**k**) Comparison of the response time and recovery time of our pressure sensor with existing pure 3D carbon-based piezoresistive sensors [4,11,12,19,20,23,24,26,27,31,32,38,52]. (**l**) Comparison of the sensitivity of our pressure sensor with existing carbon-based piezoresistive sensors. Numbers in the charts represent relevant references.

**Figure 5 nanomaterials-12-02522-f005:**
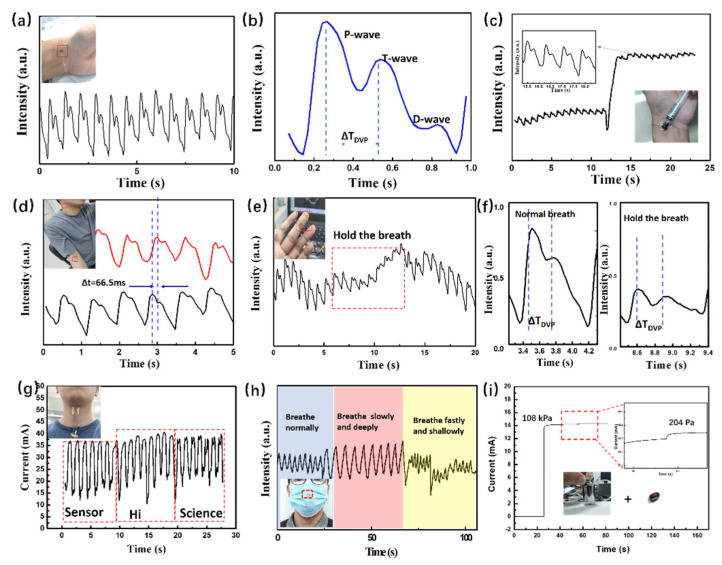
(**a**) The sensor is attached to the wrist to measure the pulse. (**b**) Pulse waves within a single period. (**c**) Pulse waves measured under external pressure. (**d**) Pulse waves from lateral epicondyle vessel of the arm (red) and carotid artery (black). (**e**) Dynamic monitoring of fingertip pulse signals under different breathing states and a single period of (**f**) normal breathing and holding the breath. (**g**) Repeated responses of pressure sensor to words with different numbers of syllables. (**h**) Real-time monitoring of human breathing with different stages. (**i**) The detection of micro-pressure caused by the load of a bean on 100 g weight.

## Data Availability

Data are contained within the article and the Appendix A.

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
