# Peer review of "Highly Sensitive Piezoresistive Pressure Sensor Based on Super-Elastic 3D Buckling Carbon Nanofibers for Human Physiological Signals’ Monitoring"

_nanomaterials, 2022, doi:10.3390/nano12152522_

Round 1
Reviewer 1 Report
The paper deals with the design and development of 14 three-dimensional buckling carbon nanofibers (3D BCNFs) for Human Physiological Signals Monitoring. The topic is certainly of interest and topicality, but the manuscript has a series of minor points, which does not make it acceptable for publication in the actual form.
1. At first the manuscript is lacking in content and the novelty of the work should be better highlighted.
2. The introduction is s too long compared to the paper's other sections. Some references about this topic may be read during the revision and might be cited in the introduction part of the revision to extend the readership.
3. Just as it is not clear from the study which the parameter of comparison with other similar systems makes it better than others.
4 The results should be compared against other published data from other studies using similar materials.
5. The conclusions do not seem to be supported sufficiently based on the data shown, an improvement is needed.
Reviewer 2 Report
The manuscript reported by Pang and co-workers is about the fabrication of super-elastic 3D buckling carbon nanofibers for human physiological signal monitoring. I found the manuscript is very well written. The experiments and results are very interesting. Thus, I strongly recommend the manuscript for publication in Nanomaterials after the authors address some minor comments as below:
1. From SEM images, the serpentine 3D BCNF is not very uniform and directional. Instead, they are randomly arranged in multi-directions. This will negatively be influenced the outcome of the sensing devices. How do the authors comment on this issue?
2. How do the shape parameters of serpentine influence the sensing performance? For example, changing the arc, width, and length of serpentine will significantly impact the stretchability of the serpentine structure.
3. The authors should provide the experimental setup for the stretchability test under applied pressure.
4. The authors should provide the stretchability test at 75kPa at different cycles to confirm the reliability of the sensing devices.
